



# Practical considerations for enhanced-resolution coil-wrapped Distributed Temperature Sensing

K. P. Hilgersom[1], T. H. M. van Emmerik[1], A. Solcerova[1], W. R. Berghuijs[2], J. S. Selker[3], N. C. van de Giesen[1]

[1]Water Resources Section, Faculty of Civil Engineering and Geosciences, Delft University of Technology, Delft, P.O. Box 5048, 2600 GA, The Netherlands
[2]Department of Civil Engineering, University of Bristol, Bristol, University Walk, BS8 1TR, United Kingdom
[3] Department of Biological and Ecological Engineering, Oregon State University, Corvallis, 116 Gilmore Hall, OR 97331, United States of America

*Correspondence to*: K. P. Hilgersom (k.p.hilgersom@tudelft.nl)

**Abstract.** Fibre optic Distributed Temperature Sensing (DTS) is widely applied in earth sciences. Many applications require a spatial resolution higher than the provided by the DTS instrument. Measurements at these higher resolutions can be achieved with a fibre optic cable helically wrapped on a cylinder. The effect of the probe construction, such as its material, shape, and diameter, on the performance has been poorly understood. In this article, we study datasets obtained from a

laboratory experiment using different cable and construction diameters, and three field experiments using different construction characteristics. This study shows that the construction material, shape, diameter, and cable attachment method can have a significant influence on DTS temperature measurements. We present a qualitative and quantitative approximation of errors introduced through the choice of auxiliary construction, influence of solar radiation, coil diameter, and cable attachment method. Our results provide insight into factors that influence DTS measurements, and we present a number of

solutions to minimize these errors. These practical considerations allow designers of future DTS measurement setups to improve their environmental temperature measurements.

## 1 Introduction

In recent years, Distributed Temperature Sensing (DTS) has been widely used for high-resolution temperature measurements in environmental sciences. While DTS instruments continue to provide ever-increasing spatial and temporal resolution, there

are many cases in which the spatial resolution is yet insufficient to address measurement requirements. Fortunately, a fibre optic cable can easily be wrapped to a coil. This makes DTS a very suitable technique to obtain dense temperature data along the axis perpendicular to the wrapping.

DTS employs Raman scattering of a laser in a fibre optic cable to determine the temperature along the cable. Scattered light in the fibre can have a wavelength decrease towards a temperature sensitive anti-Stokes signal or a wavelength increase

towards a relatively temperature insensitive Stokes signal. The ratio of the Stokes and anti-Stokes signal, and the travelling



time of the signal up and down the cable is used to determine the temperature at a certain point in the cable. More background information on DTS and its applications can be found in Selker et al. (2006) and Tyler et al. (2009).

Numerous studies applied DTS for temperature measurements with high spatial and temporal resolutions in hydrological and atmospheric systems, such as soils (e.g., Steele-Dunne et al., 2010; Jansen et al., 2011; Ciocca et al., 2012), streams (e.g.,

Selker et al, 2006; Vogt et al., 2010; Westhoff et al., 2007, 2011), lakes (e.g., Vercauteren et al., 2011; van Emmerik et al., 2013), and in the air (e.g., Thomas et al., 2012; Euser et al., 2014; de Jong et al., 2015).

In some cases, very high spatial resolutions are required to give insight into hydrologic or atmospheric systems. Various authors (e.g., Selker et al., 2006; Vogt et al., 2010; Suarez et al., 2011; Vercauteren et al., 2011; Arnon et al., 2014) have increased the vertical spatial resolution of DTS measurements by wrapping fibre optic cable around a solid PVC tube, a

technique sometimes called High Resolution DTS (HR-DTS), yielding spatial resolutions between 4 and 11 mm. The downside of this approach is that stress on the fibre or the characteristics of the supporting materials such as PVC tubes (in this article: *auxiliary constructions*) might also influence the temperature measurements. Such issues were recently pointed out by Arnon et al. (2014), but aside from this one observed problem, to date no comprehensive analysis has been published on how the physical design of a coil-wrapped DTS measurement setup affects temperature measurements.

Tyler et al. (2009) defines *spatial resolution* as the cable distance required to measure 90% percent of a sharp temperature jump. The spatial resolution is different from the spatial *sampling interval* by the fact that the Nyquist theorem requires at least twice the interval to reconstruct a continuous signal. In this article, we use the terms *machine resolution* and *coil resolution*. The machine resolution is equal to Tyler et al.'s (2009) definition of the spatial resolution along a linear section of fibre optic cable. The coil resolution is the effective spatial resolution of the temperature measurements equal to the

machine resolution divided by the coil's stuffing factor. The *stuffing factor* is calculated from the length $L$ of cable within one turn divided by its pitch $p$ (Fig. 1).

It has generally been assumed that the characteristics of the measurement setup, such as DTS cable diameter, coil diameter, auxiliary construction material, and coil preparation, do not influence the temperature measurements (e.g., Vercauteren et al., 2011; van Emmerik et al., 2013). However, recent papers discuss several errors in DTS temperature measurements caused by

practical considerations of the measurement setup. Arnon et al. (2014) describe a signal loss and consequent error in their temperature measurements along a helically wound cable. This loss was mostly visible in the first 100 m of cable and dissipated later on. They hypothesized that the curvature attenuates some of the most extreme modes of light. Further along the cable, the effect on the temperature disappeared.

Others have signalized the effect of heat conduction between a DTS cable and bedding material in streams (O'Donnell

Meininger and Selker, 2014), and inaccuracies in air and water temperatures due to solar radiation (Neilson et al., 2010; De Jong et al., 2015). The auxiliary constructions of coil-wrapped DTS are expected to deteriorate the measurement accuracy in a similar fashion.



This paper demonstrates the relationship between coil diameter, signal loss and temperature measurement error for specific fibre optic cables. With this, we aim to better relate the signal loss shown by Arnon et al. (2014) to coil diameter and fibre type, specifically, bend tolerance.

Solar radiation heats up the auxiliary construction, which can lead to inaccuracies in DTS temperature measurements. This paper discusses the air temperature measurements errors introduced by the tubular structure on which the cable is mounted, based on measurement and model results. The datasets were acquired using different types of auxiliary construction, which allowed an analysis of the influence of solar radiation for different constructions on DTS temperature measurements. For one of the field measurements, we employed meteorological data as input for an energy balance model (Hilgersom et al., 2015) to compare the fibre heating for a cable surrounded by air and a cable attached to a PVC tube. Sayde et al. (2015) has shown that energy balance models can function for modelling fibre temperatures.

Using DTS air temperature datasets obtained during one laboratory and three field measurement campaigns, the influence of coil diameter and coil preparation is evaluated. The aim is to provide practical insights into the effects of coil preparation, diameter and auxiliary construction, and propose some potential improvements to mitigate negative effects, which allows users of DTS to improve their measurement set-ups.

## 2 Methods and Materials

This section explains the methods used to demonstrate the influence of coil diameter (using a laboratory setup; Sect. 2.1), and the influence of solar radiation (using three field setups; Sect. 2.2).

### 2.1 Influence of coil diameters

One of the practical issues in applying coil-wrapped DTS is selecting the appropriate size of the coil. In general with sufficiently large diameter (over 0.1 m), the optical behaviour of the fibre optic is not influenced by bending, and performance is equal to that found in straight cable. In some installations, there is no restricting upper limit for the diameter of the coil, so use of such large diameter is feasible. For example, the environment of the installation offers sufficient space and the temperature barely varies in lateral direction from the HR-DTS pole, such as in a lake (Vercauteren et al., 2011) or on an open field (Euser et al., 2014). In these cases, the required coil resolution and the machine resolution determine the diameter of the coil, and there is usually no need to apply a coil with a very small diameter.

However, often there is a spatial limitation in lateral direction, for example, in the case of a narrow borehole (Vogt et al., 2012), or in case of strong lateral temperature variations at the 0.1 m scale. In these cases, the size of the coil diameter is restricted to often less than 0.05 m, in which case bend-related differential attenuation of the Stokes and Anti-Stokes backscatter is to be expected. The minimum coil diameter is determined by: (1) the required coil resolution, and (2) the maximum allowable signal loss by the bended fibre. The required coil resolution is dependent on the coil width and the




machine resolution. In this paper, we focus on the effects of narrow cable bends on signal loss, which might cause temperature defects.

First, the signal decay is increasing with decreasing bending radius. This *attenuation* follows from the larger fraction of the laser signal that reaches an angle of incidence inferior to the angle of acceptance in the fibre bend (Fig. 1, modes *b* and *c*). A

low number of remaining light modes leaves larger relative errors in the Stokes over anti-Stokes ratio, reducing the accuracy of the temperature measurements.

Arnon et al. (2014) found that a laser signal entering a coil at first experiences a relatively large rates of differential attenuation in the first 100 m of fibre, then asymptotically returning to constant rates of attenuation as the signal passed this distance. In their setup, the coil was of 20 mm diameter. This outcome was hypothesized to be due to the preferential loss of

the most extreme modes of light in the Anti-Stokes (shorter wavelength) compared to Stokes frequency (Fig. 2).

Second, the altered *differential attenuation* directly affects the temperature measurements. The refractive index of light in a glass fibre is dependent on wavelength according to the Sellmeier equation. Therefore, a laser signal with a lower wavelength (anti-Stokes) has a larger angle of acceptance for internal reflection. Relative to the anti-Stokes signal, the usually abundant Stokes signal has more of its modes near the critical angle of acceptance, yielding a larger consequent loss

when a bend changes the angle of incidence. In other words, the Stokes pulse loses more of its extreme modes as it passes through a bend, despite the same angular change of the guiding fibre for both Stokes and anti-Stokes modes. The altered differential attenuation of the Stokes and anti-Stokes signals was observed by Arnon et al. (2014), and is one of the reasons why Hausner et al. (2011) and Van de Giesen et al. (2012) promote manual calibration of the differential attenuation for separate sections of the cable. Unfortunately, since the loss in this case is asymmetrical with the direction of light travel (as

may be understood by thinking about the first 100 m of cable, where the light entering the coil will experience significant differential attenuation, but the light leaving the coil will have settled down to a constant rate of attenuation), the double-ended procedures presented in the literature are not applicable to these unique defects.

Important developments in fibre optic technology over the past 10 years include significant advancements in producing fibres that are 'bend-tolerant' or even 'bend-insensitive'. These terms refer to the design of the index of refraction transitions

specifically to obtain low decay when a signal passes through a bend of such a fibre. Similar to most multimode fibres applied for Raman DTS, bend-tolerant fibres normally have a core diameter of 50 μm. However, by radially varying the fibre's refractive-index, more signal is reflected back into the core instead of lost through its surrounding cladding. One example of bend-tolerant fibres is the Corning ClearCurve™ (Briggs et al., 2012). Attenuation in this fibre is reported to be less than 2.3 dB km$^{-1}$ for an 850 nm light signal and less than 0.6 dB km$^{-1}$ for a 1300 nm signal (Corning ClearCurve, 2015).

This type of fibre was unavailable in our study. Fundamentally the same losses are expected with tight-bends even when using bend-tolerant fibres, but delayed to occurring at smaller radii (e.g., Arnon et al. (2014), who employed bend-tolerant fibre). Thus our findings should translate to all fibres, though the specific bending thresholds at which the effects are observed should be expected to differ per fibre.





The cable design encasing the fibre element can also affect the passage of light. Most notably, use of a tight-buffered jacket over a loose tube cable increases the amount of strain conveyed from the cable to the fibre in comparison to loose-tube constructions. Several types of armouring may also produce micro-bends in coiled fibres.

In a laboratory experiment, we compare a tight-buffered AFL 1.6 mm simplex cable with two thicker cables for signal loss
and differential attenuation (Table 1). The signal loss is studied as a function of coil diameter and distance from the start of the coil. Our study employs the time-averaged Stokes and anti-Stokes signals along the three different cables that are wound around PVC tubes of 125, 75, 50, 32, 25, and 16 mm diameter. The tough metal casing of the thickest cable made it impossible to properly wrap it to a coil of 25 mm diameter or smaller. We therefore omitted the 25 mm and 16 mm coils for the thickest cable. The data were taken during a 65 h measurement period in a tank of water with a uniform and almost
constant temperature. The DTS instrument was a Silixa Ultima-S (Silixa Ltd., Hertfordshire, UK).

## 2.2 Influence of radiation

Influence of the auxiliary construction on the temperature measurement is generally neglected. This assumption might not hold when the auxiliary construction used for fixing the cable has high thermal mass or different thermal properties than the measured medium. In such cases, fast changes in temperature will not be correctly reported due to the thermal inertia of the
tube. This problem is for example apparent in measurements of air temperature, where there is both rapid fluctuation and low heat content. When measuring air temperature, one must also consider the effect of radiation on the probe (e.g., Vercauteren et al., 2008; Oldroyd et al., 2013). Colour of the cable coating and direct exposure to solar radiation can have influence on the temperature measurement up to several degrees (de Jong et al., 2015), and is also relevant underwater (Neilson, 2008).

### 2.2.1 Measurement data

For the assessment of the influence of the construction material on DTS temperature measurements, we compare three datasets acquired between 2011 and 2014. All experiments used different auxiliary constructions on which the fibre optic cables were mounted. The construction types varied between an almost imperforated transparent PVC tube to an open hyperboloid PVC construction (Table 2).

*a) Imperforated PVC tube, Delft (The Netherlands).* From 25 to 30 June 2012, the temperature profile in and above a ditch in
Delft (51.996° N, 4.377° E) was measured using a Sensornet Oryx (Sensornet Ltd., Hertfordshire, UK), with a 2 m intrinsic machine resolution and 1 min temporal resolution. The cable (AFL 1.6 mm simplex 50/125, white) was wrapped around a 1.8 m long transparent PVC tube with a diameter of 0.15 m, and a wall thickness of 5 mm. Small holes (5 mm diameter) were made on four sides of the tube, every 30 cm in the vertical direction. The tube is considered as imperforated, as 99.9% of the cable was in contact with the tube. The cable was wrapped around the tube with 5 mm spacing, resulting in a 0.01 m
coil resolution (Fig. 3(a)).

*b) Perforated PVC tube, Delft (The Netherlands).* From 9 July to 7 August 2014, the temperature profile in a shallow urban pond in Delft (52.007° N, 4.375° E) was measured using a Silixa Ultima-S (Silixa Ltd., Hertfordshire, UK), with a 0.3 m



intrinsic machine resolution and set to report with 5 min temporal resolution. The cable (AFL 1.6 mm simplex 50/125, white) was wrapped around a 2.0 m long transparent PVC tube with a diameter of 0.11 m, and a wall thickness of 5 mm. The tube was perforated with 2 cm diameter openings on four sides, every 7.5 cm in the vertical direction (covering approximately 5% of the total area of the tube). From the total cable, 95.0% was in contact with the PVC tube. At heights

where the perforations are centred, 77.1% of the cable was in contact with the PVC tube. The cable was wrapped around the coil with 5 mm spacing, resulting in a 0.002 m vertical coil resolution (Fig. 3(b)).

*c) Open construction, Binaba (Ghana).* From 23 to 27 October 2011, the temperature profile in the shallow Lake Binaba (10.781° N, 0.479° W) was measured using a Sensornet Halo (Sensornet Ltd., Hertfordshire, UK), with a 4 m intrinsic machine resolution and set to 1 min temporal resolution. The cable (AFL 1.6 mm simplex 50/125, white) was wrapped

around a 1.8 m hyperboloid frame, that consisted of six PVC tubes (25 mm diameter). In these tubes, grooves were made to mount the cable easily with equal spacing. The open construction was designed to minimize radiation absorption by the construction and allow water and air to flow freely through the construction. For more details on the construction, see van Emmerik et al. (2013). Because of the open construction, only 3.1% of the cable was in contact with PVC. The cable was wrapped around the construction with 5 mm spacing, resulting in a 0.004 m coil resolution (Fig. 3(c)).

The three data sets are compared for their air temperature profile measurements above the water surface. The perforated setup was used to quantify the radiation effect on the temperature measurements through the auxiliary construction. The temperature profile was separated into (1) measurement points that were only in contact with the tube ($T_{tube}$) and (2) measurement points that were in contact with both air and the tube ($T_{air}$). The spatially averaged difference between $T_{air}$ and $T_{tube}$ was used as a measure of radiation influence. Please note that since $T_{air}$ was still partially influenced by the

construction, the real deviation between temperature at the tube and in the air might be underestimated.

### 2.2.2 Modelling the radiation effect

To validate that the difference between $T_{tube}$ and $T_{air}$ is indeed caused by solar radiation, the measured differences between $T_{air}$ and $T_{tube}$ are compared with modelled fibre temperatures from a energy balance model of the cable (Hilgersom et al., 2015). The 1D model describes heat transport around the cable centre and has an equidistant grid spacing of 12.5 µm.

Incoming short-wave radiation, emitted long-wave radiation, and wind cooling calculated from the hot wire anemometer principle are source terms in the outer two cells. The modelled cable consists of four layers with properties described in Table 3. Note that several properties are assumed for our cable, and the model only serves as a general verification for our data.

Two situations are modelled: (1) a cable surrounded by air, and (2) a cable attached to the PVC tube. Because the 1D

axisymmetric model does not allow modelling the PVC tube at only one side of the cable, the following approximation is used as a proxy for Situation 2: the cable is modelled fully surrounded by a 5 mm layer of PVC, which represents the heat storage capacity of the PVC tube; afterwards, the representative fibre temperatures for Situation 2 are calculated by a





weighted average of one quarter of the modelled fibre temperatures within the PVC layer, and three quarters of the modelled fibre temperatures for the cable in air (i.e., Situation 1).

## 3 Results and Discussion

### 3.1 Influence of coil diameters

The Stokes and anti-Stokes data from the three cables in the laboratory setup were averaged over the 65 h period to reduce the effect of noise (Fig. 4), and signal loss per meter for the different coils and cables was computed (Fig. 5). In Fig. 4, bending-induced losses are characterized by a relatively large signal loss at the entrance of the coil. For the 1.6 mm and the 3 mm cables, the signal loss increases when a cable is wound around a smaller tube. Only for the 6 mm cable, the 125 mm diameter coil seems to experience a larger loss compared to smaller coils. A possible explanation is that micro-bends as a

consequence of improper winding dominate the signal loss in this case.

The bottom right pane of Fig. 5 shows the relation between the Stokes to anti-Stokes ratios and the coil diameter. Although the data is not consistent in all cases, Stokes losses appear higher when compared to anti-Stokes losses for coils with narrower bends.

This coil-induced differential attenuation was also found by Arnon et al. (2014). To complement their discussion, we

consider the added Stokes emissions in more detail. Stokes photons arise from Raman scattering. As they move on, further scattering generates secondary Stokes emissions, part of which are in modes with a near critical angle of incidence. The consequent decay of Stokes signal explains the observed exponential pattern according to Beer´s law: Stokes photons are significantly more susceptible to loss when returning to the DTS instrument from further along the coil.

To investigate the effect of the slowly dissipating signal decay described by Arnon et al. (2014), Fig. 6 separates the signal

losses of Fig. 5 into three bins that represent three sections of equal length. In most cases, the first section after the signal enters the coil shows the largest decay. The second sections show already less decay in most cases and the same holds for the third sections.

Arnon et al. (2014) demonstrated increased signal decay along the first 100 m of a coiled sensor which employed bend insensitive fibre with a 26 mm wrapping diameter. Figure 6 shows that the largest variation in signal decay can be found in

the first few meters of the coil. However, for smaller diameter coils, further signal decay follows a Beer's law exponential pattern with a decay-coefficient of about 30 m.

The wider coils show a higher variability in signal decay (Fig. 6), which can be explained by the fact that the decay is more affected by micro-bends than bending radii. After about 10 m of cable, the tighter coils still experience an increased decay as compared to the largest-radius coils. Recently introduced fibres, such as the bend-insensitive Corning ClearCurve™ fibre,

are claimed to experience less influence from bends in the cable. However, the demonstrated effects play a role in all fibre optic cables.



It should be noted that the sections in our measurement are just slightly more than 3 m long, but this offers useful insight into the physics that play a role in the first part of a cable coil. The presented losses cannot be considered representative for coils that extend over long distances. Our aim here is to show that selection of coil diameters below 32 mm, for the fibres and cable construction employed here, leads to significant losses at the start of the coil. This effect imposes a lower limit to the

coil diameter one can apply. Care should be taken when choosing both the coil diameter and the number of separate coils applied to one cable.

### 3.2 Influence of radiation

### 3.2.1 Measurement data

Figure 7 shows typical vertical profiles of air temperature at 12 P.M. during a clear day. Figure 7(a) shows a relatively

smooth profile for the imperforated PVC tube. For the perforated profile (Fig. 7(b)), one can see a clear pattern in the vertical profile. A drop in temperature was observed about every 7 cm, corresponding with the locations of the holes. This profile demonstrates the temperature difference between cable that is only in contact with the tube, and cable that is in contact with both air and the tube. In the case of the open construction, where only 3.1% of the cable was in contact with the PVC, the temperature profile reflects mainly air temperature and direct radiation (Fig. 7(c)). Due to very high sampling resolution of

measurements, and the high precision of the temperature measurements (0.01°C) in Fig. 7(b), we can also observe the influence of direct solar radiation exposure (the shaded versus the exposed side of the column) as smaller (up to 0.2 °C) fluctuations in temperature. A clear influence of incoming shortwave radiation on the temperature data is visible for our various measurement setups, especially those with PVC tubes employed as the basis of construction. The profiles in Fig. 7(a) and (c) look similar at first. However, because of the differences in construction, the temperatures were influenced by

different processes, of which the effect is demonstrated in Fig. 7(b). The PVC tube influences the heat transfer processes from air and radiation to the cable, which causes a deviation between the cable temperature and actual air temperature. The temperature measured by the cable that was placed over the perforations was up to 0.5 °C lower than the temperature measured by cable that was glued to the construction (note that all probes used similar hard PVC glue to attach the cable).

Note that the machine resolution is 0.3 m, and the perforations are 0.02 m in diameter. The temperature that was assumed as

air temperature is therefore still influenced by the tube, and hence the temperature difference between the cable in the air and attached to the tube is underestimated. Similar patterns can be seen throughout the whole measurement period. Our method to determine $T_{air}$ is a conservative estimation, and the real effect on the DTS temperature might be even higher.

A relatively open construction (i.e., a low mass density) instead of a 5% perforated tube significantly reduces the radiation effects (Fig. 7(c)). In this case, it is important to allow air/fluid circulation to prevent delayed temperature signals, which are

likely to cause delayed response, and thus hysteresis patterns.

The spatial pattern in difference between $T_{air}$ and $T_{tube}$ provides a measure of radiation influence through the auxiliary construction, reaching differences up to 0.4 °C (Fig. 8). An interesting situation occurred when fog was observed in the



morning of 20 July 2014. In this case, air temperature is higher than the temperature at the tube, reaching up to 0.7 °C. We will further omit the measurements for this particular morning in our analysis as being atypical.

### 3.2.2 Modelling the radiation effect

With an energy balance approach, we modelled the heating of the cable due to solar radiation (see Sect. 2.2). The complete
modelling results are available from Hilgersom et al. (2015). The difference in temperature between PVC and the cable is presented in Fig. 8 (blue line). From 11 to 15 June 2014, the model shows a good fit with the measurements. From 16 to 24 June 2014, the model overestimates the temperature difference. This period was relatively cloudy, compared to the first period. The model that was used to simulate the heating was not able to capture the effect of clouds. In general however, we see that the temperature difference in the cable can be explained by the effect of direct radiation. Deviations up to 0.7 °C
were measured, which introduces a relatively large error in the temperature measurements.

Figure 9(a) shows the relation between solar radiation and difference in temperature measured over the perforations and temperature measured on the tube. Different colours depict wind speed. For points with no (or very low) effect of wind (dark blue), we observe a relation between temperature difference and radiation. The temperature difference is smaller during the night than it is during the day. Presence of wind makes the influence of the tube on the DTS measurement less predictable.
With winds higher than 0.5 m s$^{-1}$, the influence on temperature varies from 0 to 0.4 °C without any relation to radiation. The only exception is night (radiation equals 0 W m$^{-2}$). At night, all temperature differences scatter around zero and even reach slightly positive values (up to 0.1 °C).

More complex patterns occur during morning and evening transitions, thus between 0 and 500 W m$^{-2}$. During clear days, with almost no cloud cover, we observed hysteresis between morning and evening behaviour of the temperature differences.
Figure 9(b) shows the temperature differences on 14 and 18 July 2014. For a sunny day (18 July 2014), the temperature difference reaches relatively high values already early in the morning, when radiation is still quite low (0.25 °C difference by 50 W m$^{-2}$). In the evening of the same day, $T_{air}$ and $T_{tube}$ reach the same values already with radiation values around 500 W m$^{-2}$. Similar, but less pronounced, behaviour can be observed also for more cloudy days (14 July 2014).

### 4 Outlook

We have shown that coil-wrapped DTS measurements involve radiation effects and light attenuation in the bended fibre. Following up on these issues, this section discusses considerations for the design, material and colour of the auxiliary construction, as well as the coil diameter.

### 4.1 Influence of coil diameters

The coil diameter influences measurements in three ways. First, a smaller coil diameter exacerbates signal decay (Fig. 4).
Smaller coil diameters significantly decrease potential measurement distance for a given cable.



Second, the decay is largest at the entrance of the coil and decreases further along the coil (Fig. 6). These results confirm the work of Arnon et al. (2014). At the entrance of the coil, the light signal loses its most extreme modes that cannot be reflected back into the bended fibre due to their angle of incidence.

Third, coil diameter directly affects differential attenuation, and consequently temperature measurements themselves (Fig.

5). To this end, we recommend careful calibration of separate cable sections to achieve highly accurate temperature measurements and not using narrow coils when it can be avoided.

### 4.2 Influence of radiation

Earlier work that used PVC tubes for coil-wrapped DTS setups discussed the heating effect of auxiliary constructions due to solar radiation (e.g., Selker et al., 2006; Suarez et al., 2011; Vercauteren et al., 2011; Van Emmerik et al., 2013). With a

simple modelling approach we show that the temperature differences measured in cable, can be attributed to the difference in heating of the cable and the PVC construction. Although our findings show that solar radiation causes temperature deviations up to 0.7 °C, transparent or light coloured PVC is still the best choice for minimizing the radiation effect. It is advisable to use a radiation model prior actual measurements to estimate possible effect of radiation driven heating on temperature measurements. Similarly, discolouring of the cables and construction by algae growth or environmental depositions likely

increases temperature errors. In outdoor applications it is advised to apply ecologically sound anti-fouling paint to prevent discolouring of the cable.

One might also employ auxiliary construction design that shades the fibre wrapped around it, such as included in the installation of Vercauteren et al. (2008). Making setups like these from PVC tubes, however, is more complicated. The emerging 3D printing technique may make such approaches more feasible.

### 4.3 Influence of coil preparation

Significant practical issues arise when building a coil-wrapped DTS setup. When a spaced cable winding is chosen, we recommend not fixing the cable into place with PVC glue or similar. These glues may be stronger than the cable's jacket, leading to tearing of the jacket, and facilitate the transmission of strain to the fibre which can cause time-varying light losses as the system heats, cools, or is deformed by environmental forcing (Fig. 10(a)). Allowing the cable to move independently

with any deformations in the supporting cylinder is preferred.

When the cable is not fixed to the support tube, a constant vertical position can be maintained by winding the cable about the tube with no space between wraps (as employed by Selker et al., 2006), or by laying the cable into a pre-formed groove that has been machined into the support tube (Fig. 10(b)). Cutting helical grooves into plastic cylinders is easily done with a gear-head lathe, and allows the builder to vary in pitch (i.e., fibre length per unit cylinder length), while keeping the cable

securely in place. Another option to fix a cable to the auxiliary construction is shrink-wrapping the coil with white plastic, which can also provide shade for the cable and lower the effect of direct exposure to solar radiation. Shrink-wrapping has been successfully applied by Suárez et al. (2011). This approach has been found successful, but may again lock the cable to





particular locations on the pole, and thus encourage transfer of strain to the fibre if the pole is bent or exposed to asymmetrical solar heating.

There are other solutions for high-resolution DTS that do not require the user to wind a fibre optic cable around an auxiliary construction. For example, Arnon et al. (2014) employed commercially prepared BRUsens temperature 70 °C high

resolution cable with a fibre stuffing of a factor 11 (fibre length per cable length). Amongst other potential setups to reach a high resolution, one could for example think of cable traverses at multiple levels (Sebok, 2013), or layered nets of cables, with the traverses or layers spaced at the demanded resolution. Such setup has the advantage that it also provides temperature data in the other spatial dimensions.

### 5 Conclusions

This paper demonstrates and suggests solutions for three practical issues involving coil-wrapped DTS: (1) influence of coil diameter, (2) impact of auxiliary construction, and (3) attachment method of the cable to the construction.

Laboratory measurements display the effect of coil-wrapped cables on both signal decay and differential attenuation. Differential attenuation affects temperature measurement and requires consideration during calibration of the wound cable sections. It was observed that especially at the start of the coil, the signal loss increased significantly for smaller coil

diameters. Increasing the coil diameter reduces adverse effects of a bended cable, although it requires more space and averages out laterally varying temperature signals.

Moreover, our data and model results show that using PVC auxiliary constructions with a high plastic mass density can cause temperature measurement deviations up to 0.7 °C. This can be even higher for other set-ups and conditions. Daily temperature deviations show a clear hysteresis pattern during clear days. The slow warm-up of the PVC cools the cable in

the morning, and the heat-retaining PVC warms the cable in the afternoon.

This paper contributes to a better understanding of the effect of auxiliary constructions on coil-wrapped DTS measurements, allowing improved designs for future measurement setups. The proposed solutions and ideas can mitigate the adverse effects of coil-wrapped set-ups on high-resolution DTS.

### Author Contributions

KH, TvE, AS designed the study and conducted the analyses; all authors contributed to interpretations and writing the paper.

### Data availability

The model data is accessible on doi:10.4121/uuid:a946eca5-0901-4a09-a95a-0c028a6b1853. All measurement data can be requested from the authors.



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

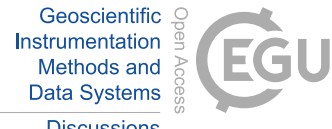




**Tables**

**Table 1.** Fibre optic cables used in this study.

| Cable | Manufacturer | Part number | Specifications | Diameter |
|---|---|---|---|---|
| A | AFL | SR0015161001 | Tight-buffered, aramid yarn | 1.6 mm |
| B | AFL | SA0015301601 | Tight-buffered, aramid yarn | 3 mm |
| C | Kaiphone | JE-2-E2000/APC-2-E2000/APC-G1-P-60-W-1500M | Loose tube, gel filled, armouring (steel flexible tube and braiding) | 6 mm |



**Table 2.** Details of the data from three fieldwork experiments in small reservoirs used in this study: (**a**) Delft (The Netherlands) using an imperforated construction, (**b**) Delft (The Netherlands) using a perforated construction, and (**c**) Binaba (Ghana) using an open construction.

| Site | Measurement period | Sensor | Machine resolution | Coil resolution | Temporal resolution | Construction type |
|------|--------------------|--------|---------------------|------------------|----------------------|--------------------|
| Delft | 25 – 30 June 2012 | Sensornet Oryx | 2 m | 0.02 m | 1 min | Imperforated |
| Delft | 9 July – 7 August 2014 | Silixa Ultima-S | 0.3 m | 0.004 m | 5 min | Perforated |
| Binaba | 23 - 27 October 2011 | Sensornet Halo | 4 m | 0.008 m | 1 min | Open |





**Table 3.** Details of the modelled tight buffered AFL cable. The layer representing the PVC tube was added in Situation 2. In Situations 1 and 2, an air layer of one cell with measured air temperature was added on the outside.

| Layer | Material | Outer radius (mm) (relative to fibre core) | Density (kg m$^{-3}$) | Thermal conductivity (W m$^{-1}$ K$^{-1}$) | Heat capacity (J K$^{-1}$ kg$^{-1}$) |
|---|---|---|---|---|---|
| Fibre & cladding | Glass | $6.3 \cdot 10^{-2}$ | $1.5 \cdot 10^{3}$ | 1.3 | $8.0 \cdot 10^{2}$ |
| Buffer | Plastic | $4.5 \cdot 10^{-1}$ | $9.5 \cdot 10^{2}$ | $3.3 \cdot 10^{-1}$ | $1.7 \cdot 10^{3}$ |
| Kevlar protection | Kevlar | $6.0 \cdot 10^{-1}$ | $1.4 \cdot 10^{3}$ | 1.2 | $4.0 \cdot 10^{2}$ |
| Jacket | PVC | $8.0 \cdot 10^{-1}$ | $1.1 \cdot 10^{3}$ | $1.9 \cdot 10^{-1}$ | $1.2 \cdot 10^{3}$ |
| *PVC tube (Situation 2)* | *PVC* | *5.8* | *$1.2 \cdot 10^{3}$* | *$1.9 \cdot 10^{-1}$* | *$1.0 \cdot 10^{3}$* |



**Figures**

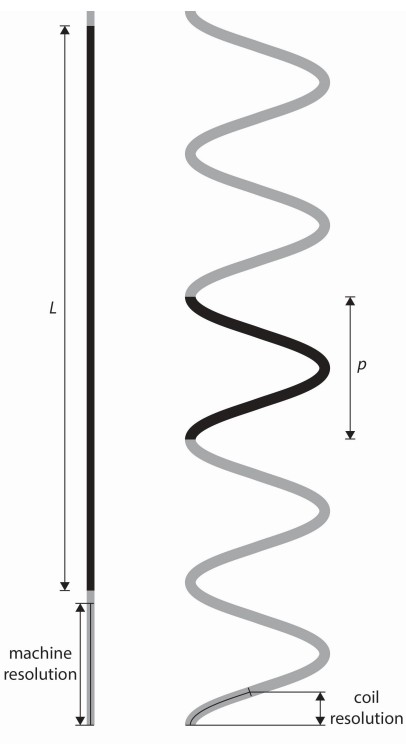

**Figure 1.** Graphical representation of used terms (machine resolution, coil resolution, and the stuffing factor defined as length $L$ over pitch $p$).



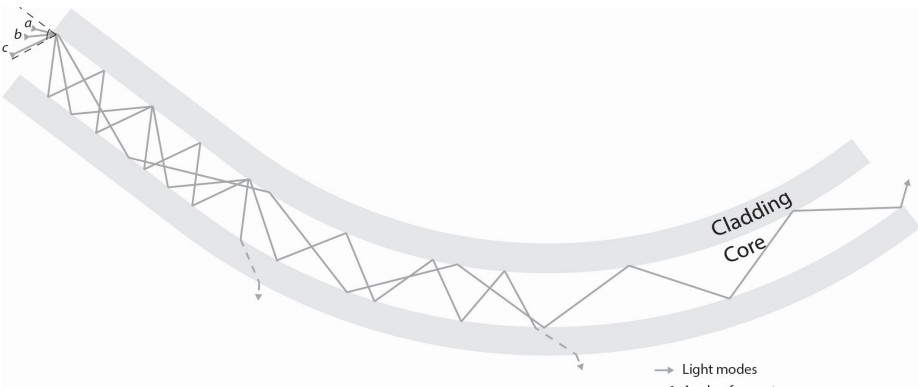

**Figure 2.** Three modes of backscattered light in a bent fibre. Mode *a* has a non-critical angle of incidence, and remains in the core of the fibre. Mode *b* falls below the angle of acceptance at a certain point in the bend. Mode *c* falls below the angle of acceptance as soon as it reaches the bend. Modes with a lower angle of incidence than mode *c* are lost as soon as they enter

5    the cable. Each angle is more acute for the higher frequency anti-Stokes backscatter, leading to a great loss of these photons than the Stokes photons, resulting in a spatially distributed differential attenuation spanning 100 m of fibre.

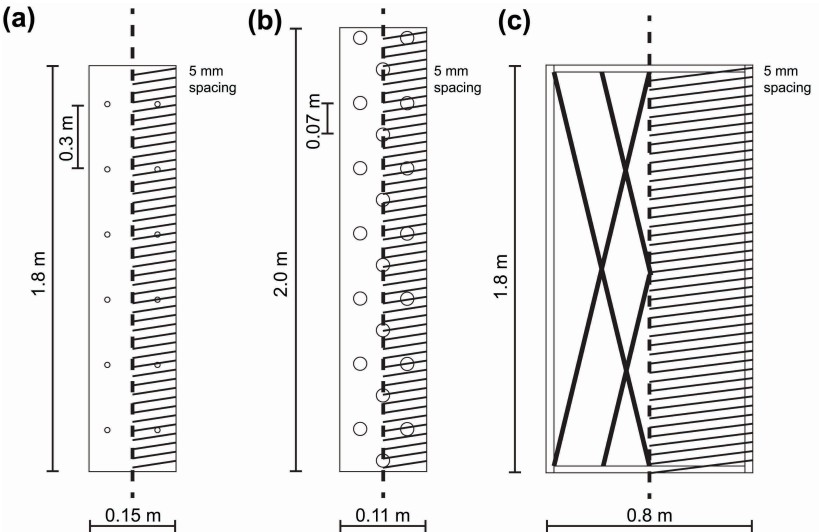

**Figure 3.** Measurement setups of the three experiments, (**a**) in Delft from 25 – 30 June 2012, (**b**) in Delft from 9 July – 7 August 2014, and (**c**) 23 – 27 October 2011.



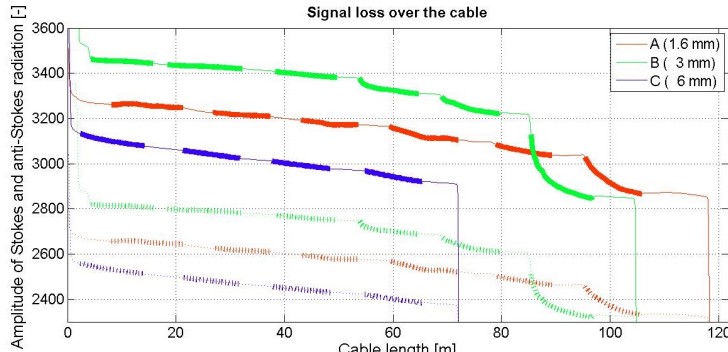

**Figure 4.** Amplitude of Stokes (solid lines) and anti-Stokes (dashed lines) radiation along the cables A (1.6 mm; red), B (3 mm; green), and C (6 mm; blue) when passing subsequent coils, marked by the thick sections of the graph. The coil diameters, from left to right, are 125, 75, 50, 32, 25, and 16 mm.

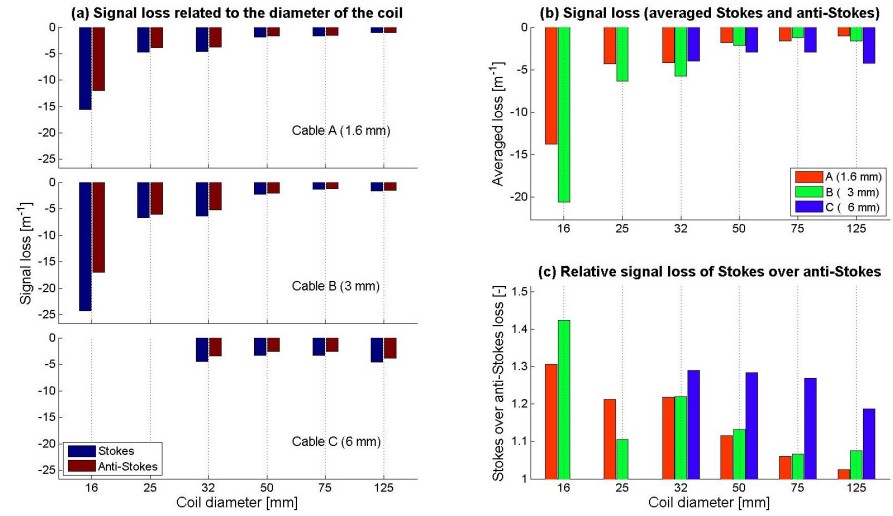

**Figure 5. (a)** Stokes and anti-Stokes signal loss (m-1) for coil diameters of 16, 25, 32, 50, 75, and 125 mm, and for the cables A, B, and C (from top to bottom). **(b)** Combination of the bar plots in the left pane using the average of the Stokes and anti-Stokes signals. Loss decreasing with increasing diameter is indicative of bend-related light loss, which is generally associated with differential loss as a function of light frequency **(c)** Ratio of Stokes over anti-Stokes losses as a function of coil diameter.




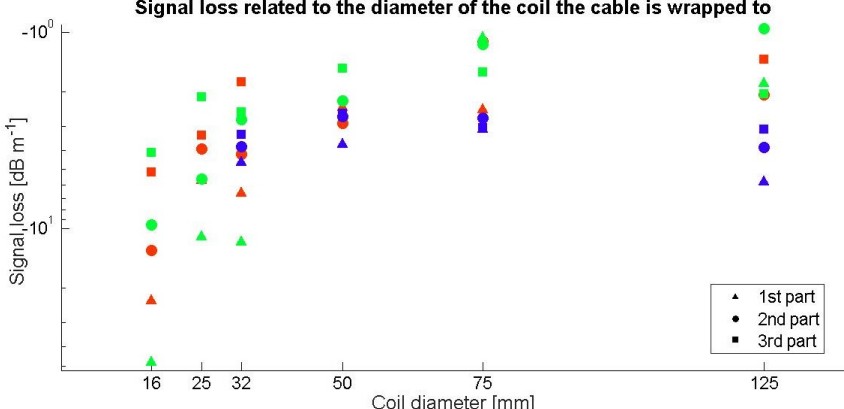

**Figure 6.** Signal loss (m$^{-1}$) over three subsequent sections of each coil presented for coil diameters of 16, 25, 32, 50, 75, and 125 mm, and for the cables A (blue), B (green), and C (red). The first, second, and third part each represent a third of the total coil length.

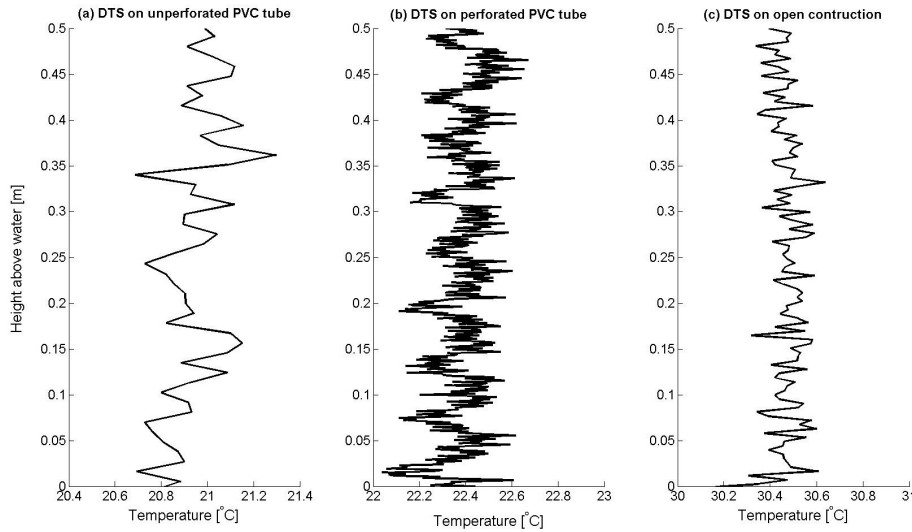

**Figure 7.** Typical air temperature profiles measured by DTS on (**a**) an imperforated PVC tube in Delft, (**b**) a perforated PVC tube and (**c**) an open construction. All profiles were taken on a cloudless day at 12PM.





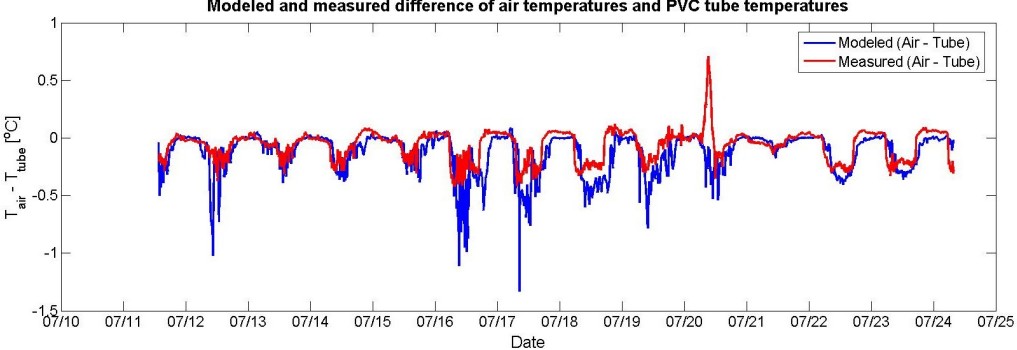

**Figure 8.** Measured (red) and modelled (blue) temperature differences between temperature measured over the holes and over PVC ($T_{air} - T_{tube}$) during July 2014 including a case of a morning dew on 20 July 2014.

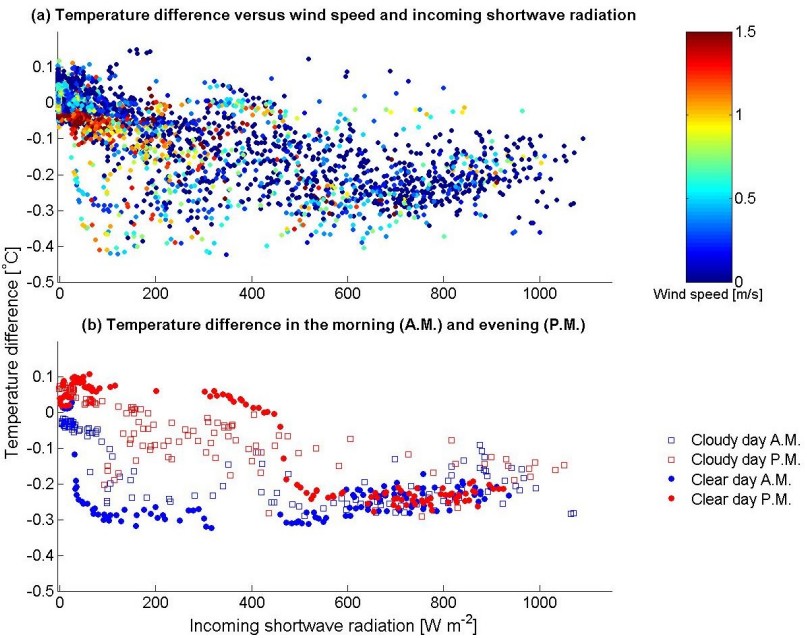

5  **Figure 9.** (**a**) Relation between temperature differences ($T_{air} - T_{tube}$), solar radiation, and wind speed (shown on colour scale in m s$^{-1}$), (**b**) hysteresis in morning and evening transition in relation between temperature differences ($T_{air} - T_{tube}$) and solar radiation.





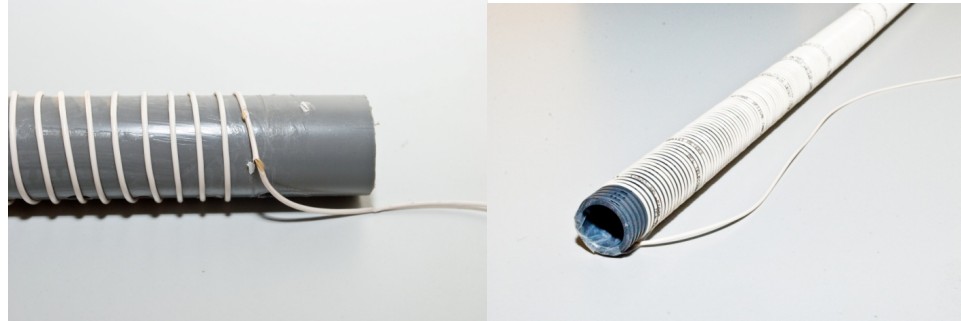

**Figure 10.** A damaged cable that was glued to a PVC conduit (left) and a cable wound through a furrow in the PVC conduit (right).