# Peer review of "Practical considerations for enhanced-resolution coil-wrapped Distributed Temperature Sensing"

_Geoscientific Instrumentation, Methods and Data Systems, 2016_

## Referee Comment (RC1) · Anonymous Referee #1 · 23 Mar 2016

**general comments The paper entitled "Pratical considerations for enhanced-resolution coil-wrapped distributed temperature sensing" focus on practical issues related to coil-wrapped Distributed Temperature Sensing. For informed experimenter of DTS measurements, it is well known (many publications can be found) that the accuracy of temperature measurements can be decreased by several effects: non uniform differential attenuation between Stokes and Anti-Stokes signals along a coil, non uniform strength reduction of signal for coil of small diameters, solar radiation. The interest of the paper is to make a wakeup call of these effects by the analysis of datasets of real experiments.**

Note: a poster entitled "Practical considerations for coil-wrapped Distributed Temperature Sensing setups" was already presented at EGU2015 (session HS1.1: Innovative techniques and unintended use of measurement equipment)

[Figure]

In the paper, there is no mention of the DTS system calibration used for the experiments. Did you use the manufacturer-internal calibration? And what are the configurations of the measurements for each experiments : single-ended or doubled-ended measurements? The double-ended measurements accounts for spatial variation in the differential attenuation of the anti-Stokes and Stokes signals. Did you try to use a manual configuration? Is it applicable for fiber optic wrapped on a cylinder of small diameter? Are there criteria of diameters and length of fiber optic wrapped on a cylinder for which the manual configuration of the double-measurement can not be applicable?

**specific comments**

- in part 1 introduction and §5 : The figure 1 is not useful. On the other hand, readers might appreciate to find : a short summary of the definition of the spatial resolution published by Tyler and al. and basic equations for the coil resolution and the stuffing factor

- in part 2.1 influence of coil diameters and §5 : As underlined in the paper, the differential attenuation between Anti-Stokes and Stokes signals could be important in case of narrow cable bends. For readers unfamiliar with guiding properties of fiber optic, it might be useful to develop the paragraph §5 ("second, the altered differentiation ..."). Moreover, the sentence "the usually abundant Stokes signal has more of its modes near the critical angle of acceptance ..." should be reformulated.

- in part 2.1 influence of coil diameters and §5 : Could you explain in more details why the double-ended procedures are not applicable to evaluate the differential attenuation along the fiber path in the case of narrow bends? Does it gives too noisy results?

- in part 2.2.2 Modelling the radiation effect and §2 : A figure of the model for the situation 1 and 2

- in part 3.1 Influence of coil diameters : the analysis of the measurements are not well presented. Perhaps, this section should be rewritten to clearly show the 3 results given

in the part 4.1

- in part 3.1 Influence of coil diameters : at the end of the part 3.1, readers might appreciate to find a short summary of the results (create a subsection 3.1.1 "Analysis of measurement data", move part 4.1 into 3.1.1 and move also part 4.3 after $3.1.1 as $3.1.2)

- in part 3.2 Influence of radiation : change part 4.2 into 3.2.3

- in part 3.2.2 Modelling the radiation effect: at the end of the $3.2.2, readers might appreciate to find a explanation of the hysteresis pattern like the one mentioned in conclusions

**technical corrections**

+Unclear sentences : - in part 1 introduction and §2 : "scattered light in fibre can have a wavelength decrease towards a temperature sensitive anti-Stokes signal or a wavelength increase towards a relatively temperature insensitive Stokes signal"

- in part 2.1 influence of coil diameters and §2 : "In this paper, we focus on the effects of narrow cables bends on signal loss, ..."

- in part 2.1 influence of coil diameters and §3 : "This attenuation follows from the larger fraction of .... . A lower number of remaining light modes leaves larger ..."

+Typing errors:

- In the title of figure 5 : "(m-1)" -1 in superscript

───────────────────────────

---

## Referee Comment (RC2) · Anonymous Referee #2 · 25 Mar 2016

The paper is interesting and give important informations on temperature measurement errors caused by the measurement setup used in high resolution DTS applications. However, there are some questions that should be clarified:

1) The Authors should be better explain how they separated the temperature profile into (1) measurement points that were only in contact with the tube ($T_{tube}$) and (2) measurement points that were in contact with both air and the tube ($T_{air}$). It unclear how this could be done taking into account that there is a very large difference between machine resolution and the perforation diameter (e.g. machine resolution is 0.3 m, and the perforations are 0.02 m). According our experience with a so large difference between the resolution and the perturbation is very difficult obtain quantitative data. Correctly, the Authors affirm that this results in an underestimation of the temperature difference between the cable in the air and attached to the tube. However, this a crucial param-

eters that strongly influence the measurements errors. The Authors should clarify if have compared the Tair, measured with the tube, with actual air temperature.

2) The Authors should clarify if the air temperature used in the energy balance model of the cable is the actual air temperature or the Tair, measured with the tube, and the influence on measurement errors

3) Have the Authors performed a comparison between the thermal inertia of the three tubes? As underlined by the Authors fast changes in temperature will not be correctly reported due to the thermal inertia especially for measurements of air temperature.

4) Measurement setups of the three experiments reported if fig.3 should be better depicted. Additional pictures could strongly the better understand the different auxiliary constructions on which the fibre optic cables were mounted.

---

## Author Comment (AC1) · 19 Apr 2016

**Referee 1 comments and answers**

We would like to thank Anonymous Referee #1 for reading the manuscript and his/her constructive comments. In the following, the questions and comments are shown in *italic*, and our replies are in **bold**.

**General comments**

1. *The paper entitled "Practical considerations for enhanced resolution coil-wrapped distributed temperature sensing" focus on practical issues related to coil-wrapped Distributed Temperature Sensing. For informed experimenter of DTS measurements, it is well known (many publications can be found) that the accuracy of temperature measurements can be decreased by several effects: non uniform differential attenuation between Stokes and Anti-Stokes signals along a coil, non-uniform strength reduction of signal for coil of small diameters, solar radiation. The interest of the paper is to make a wakeup call of these effects by the analysis of datasets of real experiments. Note: a poster entitled "Practical considerations for coil-wrapped Distributed Temperature Sensing setups" was already presented at EGU2015 (session HS1.1: Innovative techniques and unintended use of measurement equipment)*

   **The reviewer is right that non-uniform differential attenuation effects are known to scientists who are familiar with the physics of distributed temperature sensing (DTS). Over the past years, DTS has become increasingly popular as an easy-to-apply sensing tool in Environmental Sciences, and also the number of enhanced-resolution DTS applications has grown. Therefore, this paper serves to share our findings about the adverse effects that coil-wrapped setups can have on the quality of data with environmental scientists. Likewise, we demonstrate that the cable fixation method and the auxiliary construction can influence the data. Our EGU2015 poster had the same purpose. However, this article supports our message with more data, and is expected to reach a wider audience.**

2. *In the paper, there is no mention of the DTS system calibration used for the experiments. Did you use the manufacturer-internal calibration? And what are the configurations of the measurements for each experiments : single-ended or doubled-ended measurements? The double-ended measurements accounts for spatial variation in the differential attenuation of the anti-Stokes and Stokes signals. Did you try to use a manual configuration? Is it applicable for fiber optic wrapped on a cylinder of small diameter? Are there criteria of diameters and length of fiber optic wrapped on a cylinder for which the manual configuration of the double-measurement cannot be applicable?*

   **We agree that the calibration method is relevant for the presented temperature data in Section 3.2. The three field experiments all used double-ended calibration. This has been clarified in section 2.2.1.:**

**Page 5, lines 24 – 46: "For all field experiments, double-ended calibration was used, which allows for correction of Stokes and Anti-Stokes attenuation (van de Giesen et al., 2012)."**

**The double-ended calibration is also applicable to fiber optic cables attached to coils with a small diameter, because it corrects for the attenuation of the Stokes and Anti-Stokes.**

**Specific comments**

1. *In part 1 introduction and §5 : The figure 1 is not useful. On the other hand, readers might appreciate to find a short summary of the definition of the spatial resolution published by Tyler and al. and basic equations for the coil resolution and the stuffing factor §5 of Section 1 serves to clearly distinguish between the definitions of spatial resolution and sampling interval, and between the definitions of machine resolution and coil resolution, which are very relevant for enhanced-resolution DTS setups. The first two lines of this paragraph define the spatial resolution and sampling interval according to Tyler et al. (2009), and are accompanied with a reference to their article. In our opinion, these first two lines present the essence of the definitions, which fulfills for our use of the terms in this article. In case the reader wishes to read an extensive and very clear explanation of the spatial resolution and sampling interval, we recommend reading Tyler et al. (2009).*

   **We agree that Figure 1 is a little simplistic. On the other hand, it also serves as a visual introduction to the article's main topic: coil-wrapped fiber-optic cables that enhance the vertical resolution of DTS measurements. We think that Figure 1 has added value by visually introducing the topic of our article, and the terms 'machine resolution' and 'coil resolution'.**

2. *In part 2.1 influence of coil diameters and §5 : As underlined in the paper, the differential attenuation between Anti-Stokes and Stokes signals could be important in case of narrow cable bends. For readers unfamiliar with guiding properties of fiber optic, it might be useful to develop the paragraph §5 ("second, the altered differentiation ...). Moreover, the sentence "the usually abundant Stokes signal has more of its modes near the critical angle of acceptance ..." should be reformulated.*

   **The reviewer points out correctly that the sentence "the usually abundant…" is unclear. This sentence is replaced by the following three sentences:**

   **Page 4, lines 13-17: "The Stokes-scattered light modes returning through the glass fibre usually outnumber the anti-Stokes modes. Consequently, a high number of Stokes modes travel through the fibre before entering a bend. At the same time, the Stokes modes reflect from the fibre wall with a relatively small angle due to the smaller acceptance angle for internal reflection. Due to the added angular change of the fibre at the entrance of a cable bend, a relatively high number of Stokes light modes exceeds the critical angle and leave the waveguide."**

In our opinion, these three sentences better describe how a fiber bend directly affects the differential attenuation, and help develop the paragraph as requested by the reviewer.

**3.** *In part 2.1 influence of coil diameters and §5 : Could you explain in more details why the double-ended procedures are not applicable to evaluate the differential attenuation along the fiber path in the case of narrow bends? Does it gives                              too                              noisy                              results?*

This statement does not refer to the weakness of the double-ended calibration method as such. This method is usually the most accurate way to calibrate the measurement. However, the signal's differential attenuation is strongly dependent on the distance along the coil, with a sharp change in attenuation at the entrance and a fairly constant attenuation at a 100 meter further along the coil. The double-ended calibration procedures that are presented in the literature employ a piece-wise linear fit to the cumulative differential attenuation, which in this case is highly non-linear. A piecewise linear approximation of the cumulative attenuation at the entrance of the coil is undesirable over short distances, since it is labor intensive and, more importantly, more uncertain. Accordingly, this leads to noisier results. Unfortunately no methods currently exist that take this fully into account, leaving double-ended calibration as a preferred method in many cases. For example, we applied double-ended calibration to our own fieldwork measurements that are used in this study. We have added the following sentence to the manuscript:

Page 4, line 25: "However, in absence of better calibration methods, double-ended calibration is often preferred."

**4.** *In part 2.2.2 Modelling the radiation effect and §2 : A figure of the model for the situation 1 and 2.*

We agree that a figure of the cable and the surrounding PVC tube clarifies how the two situations are modeled. We added this figure as a new Figure 4.

**5.** *In part 3.1 Influence of coil diameters : the analysis of the measurements are not well presented. Perhaps, this section should be rewritten to clearly show the 3 results given in the part 4.1.*

The referee is right that the text of Section 3.1 does not present the results so that it reflects the three main conclusions on Section 4.1. At the locations in the text where we introduce the figures, we have added an explanatory sentence that presents the main conclusion of each figure, before the remainder of the paragraph further discusses of our data. We also added headers 3.1.1 to 3.1.3 to distinguish the three main results.

**6.** *In part 3.1 Influence of coil diameters : at the end of the part 3.1, readers might appreciate to find a short summary of the results (create a subsection 3.1.1 "Analysis of measurement data", move part 4.1 into 3.1.1 and move also part 4.3 after $3.1.1 as $3.1.2)*

As described in the previous answer, we have chosen for three different subheaders in the Results Section 3.1. This way, we hope to better stress the three main conclusions that can be drawn with respect to the attenuation of the Stokes and anti-Stokes signals within the coil, and make our text better understandable. We use Outlook Section 4 to briefly present our main conclusions, which are also our main warnings for future applications of coil-wrapped DTS. Therefore, we have decided not to move these parts out of Section 4.

7. *In part 3.2 Influence of radiation : change part 4.2 into 3.2.3*

    **See reply to previous Comment 6.**

8. *In part 3.2.2 Modelling the radiation effect: at the end of the $3.2.2, readers might appreciate to find an explanation of the hysteresis pattern like the one mentioned in conclusions*

    **We agree, and added an explanation in section 3.2.2.:**

    **Page 10, lines 5-7: "Since the PVC construction and the cable have different heat capacities, the heating and cooling do not occur synchronously. The warm-up of the PVC is slower than the cable, leading to a cooling effect of the cable by the PVC in the morning. In the afternoon, the PVC warms the cable."**

**Technical corrections**

1. *Unclear sentences : - in part 1 introduction and §2 : "scattered light in fibre can have a wavelength decrease towards a temperature sensitive anti-Stokes signal or a wavelength increase towards a relatively temperature insensitive Stokes signal"*

    **Page 1, lines 28-29: Changed to: "When scattering, light can have…"**

2. *In part 2.1 influence of coil diameters and §2 : "In this paper, we focus on the effects of narrow cables bends on signal loss, ..."*

    **We find it unclear what the reviewer find unclear in this sentence. We ask the reviewer to specify what is consider unclear, so we can change it accordingly.**

3. *In part 2.1 influence of coil diameters and §3 : "This attenuation follows from the larger fraction of .... . A lower number of remaining light modes leaves larger ..."*

    **In the first of these two sentences, we referred to the wrong figure, which might have caused part of the confusion here. We meant to refer to the modes *b* and *c* in Figure 2 instead of Figure 1. This has been changed in the new manuscript. Further, it might be unclear what we mean with 'angle of acceptance'. The**

sentences are changed to: **"This *attenuation* follows from the larger fraction of the laser signal that exceeds the angle of acceptance for which light is mainly reflected from the fiber wall (Fig. 2, light modes *b* and *c*). A low number of remaining light modes leaves larger relative errors in the Stokes over anti-Stokes ratio, reducing the signal strength and consequently the accuracy of the temperature measurements."**

4. *Typing errors:*

- *In the title of figure 5 : "(m-1)" -1 in superscript*

   **Applied.**

**Referee 2 comments and answers**

We would like to thank Anonymous Referee #2 for reading the manuscript and pointing out the parts of the text that need more clarification. In the following, the questions and comments are shown in *italic*, and our replies in **bold**.

1.  *The Authors should be better explain how they separated the temperature profile into (1) measurement points that were only in contact with the tube (Ttube) and (2) measurement points that were in contact with both air and the tube (Tair). It unclear how this could be done taking into account that there is a very large difference between machine resolution and the perforation diameter (e.g. machine resolution is 0.3 m, and the perforations are 0.02*

10    *m). According our experience with a so large difference between the resolution and the perturbation is very difficult obtain quantitative data. Correctly, the Authors affirm that this results in an underestimation of the temperature difference between the cable in the air and attached to the tube. However, this a crucial parameters that strongly influence the measurements errors. The Authors should clarify if have compared the Tair, measured with the tube, with actual air temperature.*

15    **We found that the distance between the temperature peaks during morning and afternoon, when the temperature difference between the cable and the PVC tube were the largest, corresponded to the distance between the perforations in the tube. Considering the perforations having 2 cm diameter and the coil resolution being 2 mm, we have around 10 measurement points per perforation.**

    **We used these points as a proxy for air temperature. We acknowledge in section 2.2.1. that our $T_{air}$ is still**

20    **influenced by the construction, as the sampling interval is indeed larger than the size of the perforations, and that we underestimate the temperature difference.**

    **We added the following clarifying text to section 2.2.1.:**

    **Page 6, lines 23-30: "The latter [the measurement points that where in contact with both air and the tube] were determined by taking the data points located at the holes in the PVC tube. The machine resolution does**

25    **not allow for extracting the temperature of the cable at the holes only, and the temperature at these points is still influenced by the PVC tube. Therefore, the values for $T_{air}$ are an underestimation of the actual air temperature. The spatially averaged difference between $T_{air}$ and $T_{tube}$ was used as a measure of radiation influence of the auxiliary construction. Please note that both measurements ($T_{air}$ and $T_{tube}$) are still influenced by exposure of the cable to solar radiation. We do not compare $T_{tube}$ with air temperature measured at the**

30    **nearby meteostation, because we would not be able to distinguish between the effect of the auxiliary construction and effect of cable exposure to solar radiation."**

2.  *The Authors should clarify if the air temperature used in the energy balance model of the cable is the actual air temperature or the Tair, measured with the tube, and the influence on measurement errors*

**The air temperature used in the model was the actual air temperature measured with an external sensor. We added the following sentence to our model description:**

**Page 7, lines 3-5: "Direct heat conduction to the air was modelled by adding one cell on the outside with air properties and a temperature forced to the measured air temperature (Onset S-THB-M008, mounted with an Onset RS3 solar radiation shield)."**

3. *Have the Authors performed a comparison between the thermal inertia of the three tubes? As underlined by the Authors fast changes in temperature will not be correctly reported due to the thermal inertia especially for measurements of air temperature.*

**The thermal inertia of the three tubes have not been compared, as investigation of various materials of auxiliary construction was not in scope of this research. We have recognized the effect of auxiliary construction on the measurements "at 12 P.M. during a clear day" (Page 8, line 25, first sentence of Chapter 3.2.1).**

**We assess the effect of thermal inertia as follows: the imperforated and perforated tubes (Figures 3a and 3b in the previous manuscript) were made from the same PVC material with an equal wall thickness. Only the diameter was slightly different (0.11m vs. 0.15m). Therefore, we assume equal thermal inertia. For the third experiment (Figure 3c), a completely different setup was used. In this case, the cable was wrapped around an open construction, with only 3.1% of the cable being in contact with the PVC. Therefore, it is also assumed that this influence is negligible, especially compared to the two tubes (95-99.9% of the cable in contact with PVC). We have given more details about the open construction with an improved Fig. 3, to highlight that this is setup is completely different, with very little contact between the cable and the PVC.**

4. *Measurement setups of the three experiments reported if fig.3 should be better depicted. Additional pictures could strongly the better understand the different auxiliary constructions on which the fibre optic cables were mounted.*

**Figure 3 has been expanded with photos of the constructions, to show how they were applied in the field.**

**Updated manuscript with tracked changes**

From the following page, the updated manuscript is displayed with all the changes added in track changes mode.

[revised manuscript text omitted]

25   temperature profile was separated into (1) measurement points that were only in contact with the tube ($T_{tube}$) and (2) measurement points that were in contact with both air and the tube ($T_{air}$). The latter were determined by taking the data points located at the holes in the PVC tube. The machine resolution does not allow for extracting the temperature of the cable at the holes only, and the temperature at these points is still influenced by the PVC tube. Therefore, the values for $T_{air}$ are an underestimation of the actual air temperature. The spatially averaged difference between $T_{air}$ and $T_{tube}$ was used as a measure

30   of radiation influence of the auxiliary construction. Please note that both measurements ($T_{air}$ and $T_{tube}$) are still influenced by exposure of the cable to solar radiation. We do not compare $T_{tube}$ with air temperature measured at the nearby meteostation, because we would not be able to distinguish between the effect of the auxiliary construction and effect of cable exposure to solar radiation.

**2.2.2 Modelling the radiation effect**

To validate that the difference between $T_{air}$ and $T_{tube}$  is indeed caused by solar radiation, the measured differences between $T_{air}$ and $T_{tube}$ are compared with modelled fibre temperatures from a energy balance model of the cable (Hilgersom et al., 2015). The 1D model describes heat transport around the cable centre and has an equidistant grid spacing of 12.5 μm. Incoming short-wave radiation, emitted long-wave radiation, and wind cooling calculated from the hot wire anemometer principle are source terms in the outer two cells. Direct heat conduction to the air was modelled by adding one cell on the outside with air properties and a temperature forced to the measured air temperature (Onset S-THB-M008, mounted with an Onset RS3 solar radiation shield). The modelled cable consists of four layers with properties described in Table 3. Note that several properties are assumed for our cable, and the model only serves as a general verification for our data.

Two situations are modelled: (1) a cable surrounded by air, and (2) a cable attached to the PVC tube. Because the 1D axisymmetric model does not allow modelling the PVC tube at only one side of the cable, the following approximation is used as a proxy for Situation 2: the cable is modelled fully surrounded by a 5 mm layer of PVC, which represents the heat storage capacity of the PVC tube; afterwards, the representative fibre temperatures for Situation 2 are calculated by a weighted average of one quarter of the modelled fibre temperatures within the PVC layer, and three quarters of the modelled fibre temperatures for the cable in air (i.e., Situation 1).

**3 Results and Discussion**

**3.1 Influence of coil diameters**

The Stokes and anti-Stokes data from the three cables in the laboratory setup were averaged over the 65 h period to reduce the effect of noise (Fig. 5), and signal loss per meter for the different coils and cables was computed (Fig. 6). In Fig. 5, bending-induced losses are characterized by a relatively large signal loss at the entrance of the coil.

**3.1.1 Coil-induced attenuation**

The top right pane of Fig. 6 generally confirms the higher signal *attenuation* for smaller coils. 
[revised manuscript text omitted]
**) Binaba from 23 – 27 October 2011, and photos of the (d) imperforated construction in Delft, the (e) perforated construction in Delft, and (f) the open construction in Binana.

[Figure]

**Figure 4.** Schematization of the cable in the energy balance model. The outer PVC layer was only present in Situation 2, where the effect of the thermal inertia of the PVC tube was modelled.

[Figure]

**Figure 45.** Amplitude of Stokes (solid lines) and anti-Stokes (dashed lines) radiation along the cables A (1.6 mm; red), B (3 mm; green), and C (6 mm; blue) when passing subsequent coils, marked by the thick sections of the graph. The coil diameters, from left to right, are 125, 75, 50, 32, 25, and 16 mm.

[Figure]

**Figure 56.** (**a**) Stokes and anti-Stokes signal loss ($m^{-1}$) for coil diameters of 16, 25, 32, 50, 75, and 125 mm, and for the cables A, B, and C (from top to bottom). (**b**) Combination of the bar plots in the left pane using the average of the Stokes and anti-Stokes signals. Loss decreasing with increasing diameter is indicative of bend-related light loss, which is generally associated with differential loss as a function of light frequency (**c**) Ratio of Stokes over anti-Stokes losses as a function of coil diameter.

[Figure]

**Figure 67.** Signal loss (m⁻¹) over three subsequent sections of each coil presented for coil diameters of 16, 25, 32, 50, 75, and 125 mm, and for the cables A (blue), B (green), and C (red). The first, second, and third part each represent a third of the total coil length.

[Figure]

**Figure 7̶8̲.** Typical air temperature profiles measured by DTS on (**a**) an imperforated PVC tube in Delft, (**b**) a perforated PVC tube and (**c**) an open construction. All profiles were taken on a cloudless day at 12PM.

[Figure]

**Figure 8̶9̲.** Measured (red) and modelled (blue) temperature differences between temperature measured over the holes and over PVC ($T_{air} - T_{tube}$) during July 2014 including a case of a morning dew on 20 July 2014.

[Figure]

**Figure 910.** (**a**) Relation between temperature differences ($T_{air} - T_{tube}$), solar radiation, and wind speed (shown on colour scale in m s$^{-1}$), (**b**) hysteresis in morning and evening transition in relation between temperature differences ($T_{air} - T_{tube}$) and solar radiation.

[Figure]

**Figure 1011.** A damaged cable that was glued to a PVC conduit (left) and a cable wound through a furrow in the PVC conduit (right).